# Influence of Surface Texturing on the Dry Tribological Properties of Polymers in Medical Devices

**DOI:** 10.3390/polym15132858

**Published:** 2023-06-28

**Authors:** Isabela Evangelista, Dorota Wencel, Steve Beguin, Nan Zhang, Michael D. Gilchrist

**Affiliations:** 1School of Mechanical & Materials Engineering, University College Dublin, Belfield, D04 V1W8 Dublin, Ireland; isabela.daconceicaoevangel@ucdconnect.ie (I.E.); nan.zhang@ucd.ie (N.Z.); 2BD (Becton, Dickinson & Company), Blackrock Business Park, Carysfort Avenue, Blackrock, A94 H2X4 Dublin, Ireland; dorota.wencel.skoczek@bd.com (D.W.); steve.beguin@bd.com (S.B.)

**Keywords:** tribology, microstructures, friction, medical devices, textured surface

## Abstract

There is a constant need to improve patient comfort and product performance associated with the use of medical devices. Efforts to optimise the tribological characteristics of medical devices usually involve modifying existing devices without compromising their main design features and functionality. This article constitutes a state-of-the-art review of the influence of dry friction on polymeric components used in medical devices, including those having microscale surface features. Surface tribology and contact interactions are discussed, along with alternative forms of surface texturing. Evident gaps in the literature, and areas warranting future research are highlighted; these include friction involving polymer Vs polymer surfaces, information regarding which topologies and feature spacings provide the best performing textured surfaces, and design guidelines that would assist manufacturers to minimise or maximise friction under non-lubricated conditions.

## 1. Introduction

Improvements in material properties and product performance lead to technological evolution. Within the medical devices industry, functionality and safety are the subject of continuous developments, and advances in polymer engineering continue to lead to materials having enhanced properties [1,2,3], such as reduced density, enhanced strength, improved thermal behaviour, transparency, wettability, biocompatibility and chemical resistance. In many instances, polymer materials are competitive alternatives to the use of metals or ceramics [4]. The use of polymers is increasingly prevalent for tribological applications in medical devices, and this has been driven both by requirements related to product shelf life and component functionality. Zhang et al. [5] conducted a recent literature review on tribological influences associated with biomedical devices, including artificial joints, fracture fixation, and surgical instruments. What was common to the various applications was that product performance was often limited by tribological properties, which confirms that innovative approaches are required to improve contact interactions that would serve to reduce or avoid component wear and damage.

Interactions in the form of sliding contact between components in medical devices are frequently responsible for limiting the functionality and performance of a product. In artificial joints, the main failure mechanisms are wear loss and the interaction of material debris with bone and biological tissue. Since the durability and viable service-life of such prosthetic devices is only around 15 years, their use for younger patients having a longer life expectancy can be problematic [6]. Material and design solutions, as discussed in [7], can address this challenge by means of reducing wear and friction. Similarly, for non-invasive medical devices, friction and wear between mechanical components that involve sliding contact are often presented as an engineering challenge. Researchers generally regard wear and friction as a constraint to improving the relative motion between mechanical components while maintaining overall product performance. In such instances, lubricants are often used as a solution, although there are many medical device design situations which prohibit their use, as will be explored throughout this article.

A reduction in friction at the interface between a pair of moving surfaces can serve to reduce wear. Besides changing material combinations, the use of liquid or solid lubricants and modifying surface contact areas are ways to provide alternative options for reducing friction in medical devices [8,9,10,11]. Surface modification techniques can alter the texture of a contact surface through mechanical or chemical processes, and modifications can be in a form of material transfer, coatings or material loss which will affect the subsequent wear mechanisms and rates [12].

Besides allowing friction properties to be optimised, surface texturing is also used to provide antibacterial or antiviral properties to a medical device and to control the surface wettability of a medical device [13,14,15]. However, the particular focus of the present article is the texturing of surfaces that serve to alter or control tribological properties under dry contact conditions, i.e., without lubrication. In many design circumstances, the use of lubricants is often not permissible due to reasons of incompatibility, contamination, and lubricant migration, which would damage the functionality of other components of a medical device. We summarise the use of polymer materials for non-invasive medical devices, i.e., medical devices which are external to the human body, and focus particularly on tribological considerations at the sub-millimeter scale. This review focusses on polymers, but includes some aspects of metals and biomimetic texturing, and concludes by highlighting gaps in the current state-of-the-art and by suggesting important application areas that would benefit from further research.

## 2. Tribology and Medical Devices

According to the European medical device regulation, EU 2017(745), there are four main categories according to which medical devices are divided, varying from the lowest to the highest associated risk of injury or illness. Class I is the lowest risk category, e.g., wheelchairs, thermometers, glasses, hospital beds, followed by Class IIa, with a slightly higher risk, that includes devices used for disease treatment/tracking such as hearing aids, sphygmomanometers, syringes, pregnancy kits, etc. Class IIb and Class III have the highest risk categories: Class IIb includes medical devices such as dialyzers, surgical lasers, devices to prevent pregnancy and lens, while Class III devices have a direct connection to the body and include hip implants, cardiovascular pacemakers, etc. [16].

Regardless of the class of a medical device, surface interactions always influence the performance of a medical device, and such interactions between surfaces relate to the study of tribology. Biomedical applications that involve continuous mechanical motion, such as prosthetic implants, artificial joints, or moving mechanisms in a drug delivery device generally have higher failure rates due to friction and wear [17,18,19] than devices that involve no relative motion between their various components. Devices involving drug delivery, such as syringes or auto-injector pens, are also affected directly by friction mechanisms, such as exist between contacting mechanical components that are in relative motion during normal performance of the device [20,21,22]. Figure 1 illustrates typical medical device examples where non-invasive sliding contact is observed.

Kasem et al. [23] investigated whether the design of plunger surfaces would alter dynamic friction. Experimental conditions involved testing different injection liquids at three sliding velocities, 2.5, 5, and 10 mm/s, to evaluate friction performance between the Poly(vinylsiloxane) (PVS) plunger and the barrel of commercial medical syringes. Textured barrels produced by a casting process included 200 µm diameter dimples of 20 and 50 µm depth. A reduction in friction was observed when using a textured surface rather than a smooth surface, since a lower resistance force was observed, and this was true regardless of the injection liquid and sliding velocity. Greater reductions in the friction force were associated with texture designs that were dimensionally smaller. Figure 2 shows one of the textured plunger surfaces.

In an earlier investigation into reducing friction forces, Siniawski et al. [8] used lubricious films between a syringe barrel and the rubber plunger. When using L-OMCTS silicon film, the coefficient of friction reduced by 15% compared to the unlubricated conditions. Besides barrel—plunger interactions, the tribology of interactions between a needle tip and skin has also been studied [24,25] in order to establish what surface texture on needles would minimise friction during percutaneous injection procedures on the skin. Blended channel textures containing micro dimples and micro-channels were found to reduce friction. These studies suggest promising avenues to reduce friction forces via surface textures. Other studies enhancing functionalities of medical devices components can be found in [26,27,28].

However, devices such as pacemakers, wheelchairs, and dialyzers also provide significant scope for optimization of the tribological behavior since they contain mechanisms involving interactions between moving surfaces. Xie et al. [29] reviewed the tribological challenges of cardiovascular devices, particularly those pertaining to mechanical wear and friction, fluidic-friction, and friction between the cardiovascular device and soft tissue. Surface modification, redesign, and lubrication were identified as alternative ways of improving the device’s performance.

To optimise tribological properties, it is necessary first to understand application requirements such as material loading, body interactions, durability, and the mechanism of action of the medical device. For components that are made of thermoplastic materials, plastic deformation/debris, sterilization aspects, and affinity are all important factors. The polymer materials chosen for use in medical devices will depend on whether the device is used externally or internally by a person. Table 1 summarises the most prevalent polymers currently used in medical devices.

## 3. Tribology of Contacting Surfaces

### 3.1. Surface and Contact

Surface roughness pertains to the characteristics of a surface, although this differs at the macroscopic and microscopic scales. According to the Terrace Ledge Kink model (TLK model), a surface comprises several atomic layers showing different energy bonding. Interior layer atoms show higher bonding energy when compared with surface atoms. Irregularities on the topmost surface level can have the form of peaks and valleys when observed on the microscale; these are commonly named asperities [44].

The distribution, size, and shape of asperities have a significant influence on surface roughness and interactions between contacting surfaces. Typically, their distribution and shape are random, although they can be approximated using specific geometries for surface mapping. As regards the tribology of contacting surfaces, this is defined by contact between the top surface of opposing asperities. The apparent, nominal, or effective contact area is defined by the overall contacting dimensions of opposing surfaces. The real contact area is the actual area that is physically in contact. As a surface is composed of randomly distributed asperities, the measurable contact area in these features is what defines the real contact area. By way of a practical example, when a normal load is applied to two opposing surfaces, contact occurs first between asperities on each surface. As loading is increased, the real contact area approaches that of the apparent contact area. This phenomenon is described in Figure 3 and occurs due to the deformation of asperities [45].

During contact interactions, mechanical behaviour changes with different materials and geometries. Of the numerous studies on changes to mechanical behaviour due to contact, Hertz, Bowden, and Tabor, and Greenwood and Williamson were amongst the first to investigate surface contact interactions [46]. At the end of the 19th century, it was Hertz who was one of the first to study contact mechanics by describing frictionless behaviour between solids in contact, and proposed that solids in contact exhibit elastic behaviour in response to load. Later, Johnson, Kendall, and Roberts studied larger deformations during contact than those studied by Hertz, which led to the JKR theory that considered surface energy forces during contact mechanics. Bowden and Tabor also made a significant contribution to the study of contacting bodies by being the first to consider the influence of asperity contact on friction. Their experiments showed a dependence of friction with real contact area in metal-metal contacts. Greenwood and Williamson extended their experiments and explored contact between flat and rough surfaces [47].

### 3.2. Interactions during Contact

While several factors influence static and dynamic contact between two interacting surfaces, it is friction that is the focus of this present article. According to Wang et al. [48], friction is described as the force resisting relative motion between solid surfaces, fluid layers, and material elements. Depending on whether motion is actually happening or not, friction will have two forms, which are termed static friction or kinetic friction. Static friction is characterized by external forces that maintain the object at rest, until motion is pending [49], during which the friction force is directly proportional to the normal component of the force that is applied to the object, and is independent of the contact area [44]:(1)$$F_{s}=\mu_{s}N$$
where *µ_s_* is the coefficient of static friction (1). When movement occurs between surfaces, the friction force acts in the opposing direction and is smaller in magnitude than the static friction force [50].

Adhesion is also present during interactions between opposing surfaces and is considered to be a tribological force. This force can be physical or chemical, exists on a molecular scale, and its interactions are characterized as being either repulsive or attractive depending on the affinity of the contacting molecules. Van der Waals and hydrogen bonds are most dominant as interactive forces, but the influence of stress is also a part of adhesion [51].

Associated with surfaces that are in contact, wear is defined as progressive material loss due to several interactions between surfaces. Those interactions follow mechanical stress, temperature changes, and chemical interactions. Three known types of wear categorize the aforementioned interactions: abrasion, adhesion, and fatigue [52]. Abrasive wear is differentiated as either two-body abrasive wear or three-body abrasive wear. When contact occurs between a soft surface and a hard surface, abrasion is observed via material loss from the soft surface. The mechanism occurs as a result of one surface ploughing into the other and is called two-body abrasion wear. Three-body abrasion wear takes place when particles on both surfaces are in contact, regardless of whether the surfaces are smooth or rough: material removal occurs from either or both surfaces [17,53]. Adhesion wear is the other type of interaction, which occurs when opposing solid surfaces are bonded together and material transfer occurs from one surface to the other. Finally, fretting fatigue wear occurs as a result of repeated cyclical contact across the same area, and this generates debris [54].

### 3.3. Contact between Surfaces—Emergent Research

Tribological studies of polymer-polymer contacting surfaces are less extensive than those between metals or ceramics. In tribological investigations, researchers typically examine the effect that operational or process parameters have on friction or wear [55]. Chaudri et al. [56] examined the dry sliding frictional properties of Polybutylene terephthalate (PBT) blended with PTFE in contact against Polyoxymethylene (POM) using a linear reciprocating tribometer. Smaller values of the friction coefficient were associated with slower sliding speeds (10 mm/s) and higher loads (20 N). As for wear, the PBT pin showed plastic deformation at loads higher than 5 N, which was believed to be due to increased stresses and temperature during contact. Similarly, Laursen et al. [57] evaluated the tribological behaviour on Polydimethylsiloxane (PDMS) and PTFE to understand the influence of additives on injection-moulded polyacetal. Using additives to reduce friction can affect mechanical properties and processing conditions, since these also directly affect material physical properties. Overall, the friction forces were seen to reduce with the use of additives, although mixing PTFE with pure POM led to no such change. As for the varied processing parameters, they had no effect. Polymer composites, glass fibre-reinforced thermoplastics, have also been studied for friction and wear performance by Unal et al. [58], who measured the coefficient of friction by using a pin-on-disc apparatus. They found a diminishing coefficient of friction with increased load for most tested polymers. Unsurprisingly, the wear rates for Polyphenylene sulfide (PPS), PA 46, and PA 66 were higher with increased load levels. From the results observed in previous experiments, changes to the coefficient of friction with changes in normal loads could be due to changes in surface roughness arising from differences in stress and strain.

There have been significant studies on the tribology of polymer-metal contact [59,60]. Pogacnik and Kalin [61] examined the sliding wear behaviour of polymer-polymer, PA6-POM, and polymer-metal contacts, PA6-steel. In their tests, rough and flat surfaces were used, and higher friction coefficients and wear rates were observed for interactions between smooth PA6 and POM surfaces. For PA6 versus steel, an increase in the coefficient of friction occurred as a result of PA6 film being worn onto the steel surface. Evident surface changes occurred in the form of film formation of PA6 on the steel-disc, which would explain the changing coefficient of friction. These results could confirm adhesion wear in the sliding process. Finally, in terms of temperature effects during contact, higher temperatures were observed only during polymer-polymer contact as a result of the thermal conductivity of the materials. However, no testing of disc specimens that contained microfeatures on disc surfaces were carried out in order to establish whether these would alter the measured coefficients of friction. Mergler et al. [62] also compared polymer-polymer and polymer-steel sliding contacts. They observed material transfer from POM to metal, which led to an increase in the measured coefficient of friction. Polymer wear was more evident in polymer-polymer contact than in polymer-metal contact. Similarly, Endo and Marui [63] studied the tribological behaviour between POM and carbon steel. Two surface tips were used for pin-on-disk tests in order to ascertain whether specimens having different tip geometries that were in contact with flat surfaces behaved differently. Wear was more pronounced for contact between the same materials, regardless of the tip shape, and they observed no differences between specimens having different tip geometries. In work by Sudeepan et al. [64], the tribological behaviour of an Acrylonitrile butadiene styrene (ABS) polymer composite containing Zinc oxide (ZnO) filler was observed by using a block-on roller multi-tribotester. Their friction tests indicated that the coefficient of friction and wear rate were both reduced at increasing loads and speeds. The addition of filler improved the tribological properties of ABS (i.e., reduced friction and wear), but the greatest reduction in friction was associated with a 5 wt % filler content, while a 15 wt % was required to obtain the lowest wear rate. Following those experiments, future work could examine lower filler contents in order to establish what minimum amounts cause friction to decrease.

## 4. Surface Texturing

Creating textured surfaces is well recognised as a route to altering the tribological properties, including friction and material wear, of an object. Many textured designs have been inspired by biomimetic examples that are widespread throughout the world of animals and plants. Surface modification has been a significant research topic for many decades, with an early example being the results obtained by Etsion’s group, which sought to improve sealing performance by means of surface alteration [65]. While several articles have been published on the topic, it is often difficult to compare the results of different researchers, since testing methods are not always consistent. Nonetheless, a common observation in all investigations is that the coefficient of friction depends on the surface, normally applied load, and material properties. Surface texture often decreases friction, but for wear and fatigue, contrary effects are often reported. Future studies should serve to elucidate the influence that the form and shape of different surface textures have under both lubricated and dry friction conditions, in order to establish more comprehensive conclusions [66].

When providing a textured pattern at the microscale, it is preferable that the materials, as mentioned previously, should have a low friction coefficient (smaller than 0.2) and a high wear resistance, in order that the dimensional stability of the microfeatures is maintained. There are several fabrication processes that can be used to obtain microfeatures (e.g., pillars, holes, dimples, ridges, channels, grooves, etc.), and their choice depends on the material and final application of the component or product. Moulding processes are methods used for manufacturing components at relatively low cost and in high quantities, and are predominantly used with polymer materials. Micro injection moulding and micro hot embossing are the most common of these forming processes. In micro injection moulding, textured moulds are used to imprint the inverse texture onto molten materials within the mould cavity [67,68]. Song et al. [69] investigated which process parameters proved most effective in controlling the accuracy and quality of replicated micro pillar arrays using micro injection moulding.

Micro hot embossing was found to be more accurate than microinjection moulding, since the mechanical stresses induced within the polymer are smaller in magnitude. In this technique, a mould is used to imprint the inverse of a pattern by means of applying pressure and temperature to the polymer [70,71]. This can be used to create a hydrophobic surface, while other notable techniques include photolithography, which is used throughout the semiconductor industry and consists of transferring a pattern to a specific material surface using light exposure on to a mask cover film [72,73]. Similarly, nanoimprint lithography allows for smaller patterns to be produced by using a stamp that comprises the nano-patterns to be replicated [74,75].

Besides moulding and forming processes being used to create a textured surface, subtractive and additive manufacturing can also be used, common examples of which include 3D printing, sand blasting, laser surface texturing, and electric discharge machining [76,77,78,79,80,81,82,83,84].

### 4.1. Metal Texturing

Several research studies have sought to provide a surface texture on metals. For Hao et al. [85], surface texturing was achieved by patterning a 6 cm tall cylindrical surface of a rotor-bearing system. Electrochemical micromachining was used to create cylindrical features with aspect ratios of 1:2 and 1:4 on the tin-bronze surface. The micropatterning surfaces enhanced the dynamic stability of the rotor-bearing system by reducing shaft vibrations. Wei et al. [86] investigated the frictionreducing performance of laser-fabricated micro dimples that were circular in shape on bearing steel surfaces. Study dimple density was around 25, 36, and 63%. Under lubricated conditions, friction decreased more significantly than under dry conditions. The friction reduction was 4–7% under dry conditions, while it was 45.5–60.3% when lubricated. Overall, for these experiments the results indicate that friction reduction occurs with an increase in texture density. By means of simulation and corresponding experiments, the reduction of friction was explained in terms of the reduced effective contact area due to the presence of the dimples. This observation clearly suggests that a micro-dimpled texture could serve to reduce the coefficient of friction associated with an interfacial surface. Figure 4 shows textured surface before and after sliding tests.

Cho and Choi [87] also examined the extent to which micro grooves as a surface texture would affect interfacial friction. They used a pulse laser to texture AISI 1045 steel surfaces with patterns of different densities. Their testing method involved using an ultrahigh molecular weight polyethene (UHMWPE) pin against the textured surface. Results showed a decrease in the coefficient of friction from 0.26 to 0.092 when comparing samples with flat and textured surfaces. Figure 5 shows SEM images of textured samples used in these experiments.

Shimizu et al. [88] machined C2680 brass with different microfeature geometries, namely triangular, parallel, and perpendicular grooves, to evaluate tribological properties. Micro dimpled samples with 40% of area density, and the triangular shaped dimples had the lowest coefficient of friction. They also noted that friction was higher for lower density textures. AISI 52100 bearing steel was used in a similar surface texturing study by Iqbal et al. [89]. They used a ball-on-disc tester to measure the coefficient of friction of patterned microstructures that were achieved by laser surface texturing. Different dimple densities, 5% and 10%, and an untextured control surface were examined under different nominal load levels. At the highest load level, the textured surfaces exhibited a lower coefficient of friction than the nominally smooth surfaces; a 10% dimple density had the lowest value, i.e., 0.42, which was almost 20% lower than that of the 5% dimple density surface. The use of dimpled textures was compared to grooved textures by Bhaduri et al. [90] on a tungsten carbide surface, while both textures were obtained by means of laser texturing. In a series of sliding tests, both textured surfaces were seen to trap wear debris and had a lower coefficient of friction than the nominally smooth surfaces. Other investigations on metal textured surfaces and the associated functional response are discussed in [91].

### 4.2. Polymer Texturing

Korpela et al. [92,93] studied the influence of friction and wear associated with textured patterns of both POM and PP, which had been manufactured via microinjection moulding. Five different levels of surface coverage were explored using tapered cylindrical features with the PP material, illustrated in Figure 6, from which it was concluded that a 16.8% surface coverage of a microscale feature would exhibit a lower coefficient of friction when tested against steel discs of varying roughness. For wear behaviour, the same textured surface had a lower resistance when compared with a flat sample.

The same authors also examined similarly textured POM samples and compared the results against those of PP. Greater resistance to wear was observed for POM, which also exhibited a lower coefficient of friction than PP. Typical worn POM features are shown in Figure 7 [93].

Various concave circular microscale features were laser machined on a steel surface by Qi et al. [94], as shown in Figure 8, following which a PTFE/Kevlar fabric composite was attached to the surface. A reference flat surface was used for comparison purposes. The measured coefficient of friction of the various textured surfaces was approximately half that of the reference flat surface. In a series of wear tests for the same surfaces, the wear rate was faster for the flat surfaces and slower for the textured surfaces at the same test speed. The authors concluded that wear and friction properties of textured samples had an improved tribological performance when compared with flat samples, which, for this case, featured the trapping and formation of PTFE film [94].

Wang [95] also used laser machining to create dimpled cavities on Polyether ether ketone (PEEK) surfaces, and then evaluated the tribological performance during sliding contact against 304 stainless steels. Wear debris were smaller for an array of circular dimples of 50 µm diameter than 25 mm diameter dimples. However, smaller values of the coefficient of friction were observed for the 25 µm diameter dimple array, when compared with the flat samples (0.35 vs. 0.37, respectively). Further studies ought to be performed to examine this further, since the variation in the coefficient of friction was not observed for lower loads.

He et al. [96] used photolithography to fabricate moulds that were used to produce textured samples in PDMS, the features of which included arrays of square pillars and rows of parallel grooves, as shown in Figure 9. The characteristic dimensions of the features were particularly small, i.e., 10–25 um. In their subsequent series of friction tests using the square pillar arrays, the smallest measured coefficient of friction was associated with the samples having the smallest actual contact area. Unsurprisingly, for the grooved textures, the measured coefficients of friction were different when the motion was parallel or perpendicular to the alignment of the grooves. Regardless, however, the friction coefficients of the textured surfaces were smaller than that of the nominally smooth, untextured surface.

### 4.3. Biomimetic Texturing

Within nature, there are extensive examples from the world of animals and plants that exhibit special tribological characteristics in respect of minimising or maximising friction, drag, and adhesion forces in order to satisfy a particular function. Many plant surfaces combine chemical and architectural features that provide a specific ability. A recent review article by Barthlott et al. [97] describes many of the structural characteristics of plants and illustrates how these can inspire biomimetic innovations. The floating fern, Salvinia, entraps a layer of air that serves to reduce drag and friction, and to insulate the plant from water, while water is naturally repelled by the super hydrophobic surfaces of the Lotus and Colocasia flowers due to the micro-sized protuberances, as shown in Figure 10 [98,99].

There are equally significant biomimetic examples of tribological applications related to friction manifest on the skins of animals, such as snakes, lizards, and sharks [100,101,102]. Polymeric surfaces that exhibit low levels of dry friction (i.e., unlubricated) have been manufactured by Baum et al. [103] and Wang et al. [104] using surface topologies that have been inspired by the skin of snakes, as illustrated in Figure 11.

A lizard’s ability to climb walls and to adhere upside down to surfaces in wet or dry conditions is renowned. From a tribological perspective, friction and adhesion forces act according to different rules in different sliding directions, and it is these that enable lizard locomotion [105]. Figure 12 shows different scales of the foot of a gecko lizard, which illustrate the mechanism by which it has such good adhesive properties [106].

Shark skin and beetle shells are other textured surfaces that have provided inspiration for reducing turbulence, drag, and friction, and for preventing biofouling [107,108]. Riblet shapes on shark skin, illustrated in Figure 13, usually change according to species and location and are also flexible [109]. Chen et al. [110] presented an overview of methods that have been used to manufacture shark-skin inspired surfaces, while Pu et al. [111] showed the effectiveness of such surface textures in preventing algae and microorganisms from attaching to a boat’s hull.

## 5. Conclusions

Synthetic polymer materials started to appear as a replacement for metals in the early decades of the 20th century. Besides reducing weight and improving product performance, the use of polymers also helped to decrease incidences of infection associated with medical devices. Improvements in polymer materials have led to enhanced characteristics and the surface properties of medical devices. Tribological improvements have led to polymeric components and products with controlled levels of wettability, biocompatibility, antifouling, adhesion, and friction. Table 2 summarises the reported range of coefficients of friction associated with contact between various smooth and textured polymers and other materials in the absence of lubrication. This article has focused on the texturing of polymer surfaces that are in dry contact with other surfaces during the normal operation of medical devices. The use of lubrication has specifically not been included in this review. From the table, it can be seen that there are relatively fewer articles that have considered the influence of pillar-shaped microstructures than grooved geometries. Geometric shape clearly has a significant influence on friction behaviour, since the smaller effective contact area diminishes the coefficient of friction. This present review confirms that when the textured surface is metallic, the use of grooves is most common, while, for textured polymer surfaces, the use of grooves and pillars is equally common. As regards the friction reduction of textured surfaces, the percentage reduction tends to be higher when polymer materials have intrinsically smaller coefficients of friction, i.e., µ < 0.2, for example, UHMWPE 65% and PTFE 69%. When it comes to polymer surfaces that are textured, the higher reduction of 65%, is observed on PMMA, as compared to 43% for PDMS and 41% for PVS. Other reductions have been observed in textured metal surfaces, but these are notably smaller in percentage terms. Nevertheless, reductions were always observed when sliding contact was present between two surfaces where one surface was textured.

Surprisingly, to the best of our knowledge, no studies appear to have investigated friction, wear, and sliding contact between textured and non-textured polymer samples. A reason for this may be due to the fact that tribometer studies tend to be carried out using standard steel counter surfaces. Additionally, sliding contact has long been a subject of study for metallic components that are subjected to relative motion, e.g., mechanical gears.

As the extensive examples discussed in this article show, surface texture affects material behaviour, and sliding contact is made easier by decreasing the interfacial friction forces via the use of textured surfaces rather than nominally smooth, untextured surfaces that are in contact under dry conditions. Design considerations for whether or how to texture a particular surface of a product also include material, manufacturing, and environment. Commonly used texturing features include pillars and grooves, and the inverse of such shapes, i.e., holes and ridges, while variables associated with such features include topological shape, aspect ratio, physical dimensions, scale, spacing, and arrays, or combinations of such features. The essential characteristic of all such designs is that they ensure there is a reduction in the effective contact area between opposing contacting surfaces, which leads to a reduction in the associated friction forces when observed in dry environmental conditions. Besides a reduction in contact area, such features can also promote air entrapment, which can serve to generate a cushioning effect, resulting in a reduction in the amount of direct contact between surfaces; interlocking between surfaces can also be noticed.

As regards thermoplastic materials themselves, various polymers and blends of polymers are formulated to provide smaller coefficients of friction as an intrinsic material property [116].

## 6. Future Trends

Significant future work needs to be undertaken to address the current gaps in state-of-the-art knowledge that are evident from this review. Concerning design considerations for polymer interactions, new design shapes, such as positive and negative dimples and tapered features, as well as various arrays of such features need to be examined under dry sliding conditions in order to ascertain actual and theoretical levels of friction and wear, and how these compare against nominally smooth sliding surfaces. A systematic understanding of how various polymer materials with high coefficients of friction should best be textured in order to achieve a desired mechanical performance needs to be established for particular biomedical applications. In particular, there is a significant paucity of primary data on friction and wear associated with polymer versus polymer sliding surfaces, and there is a strong need to quantify the improvements in friction and wear that texturing of one surface can provide in such material combinations, particularly as there is lack of consistency between results published in different articles. Standardization of test methods and protocols, as well as thorough calibration of equipment and sample preparation procedures would serve to improve the consistency of friction coefficient measurements.

Addressing these current knowledge gaps by creating new primary data on polymer-polymer friction and wear, and by establishing design guidelines on the texturing of surfaces that serve to enhance product performance will significantly help in designing future generations of medical devices containing polymer components, minimise levels of material consumption, and improve the performance of such devices.

## Figures and Tables

**Figure 1 polymers-15-02858-f001:**
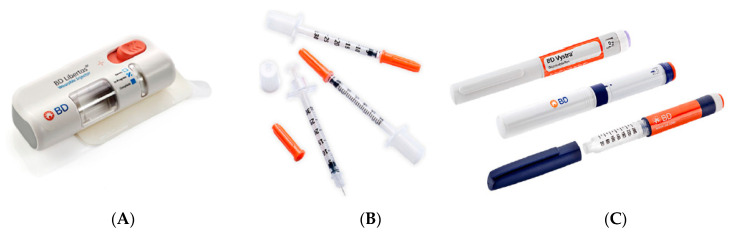
Typical drug delivery systems with mechanical components that involve dry sliding friction (**A**) wearable autoinjector (mechanical system inside of the device); (**B**) syringes (contact between plunger and barrel); and (**C**) pen-needles (mechanism of measurement dosage).

**Figure 2 polymers-15-02858-f002:**
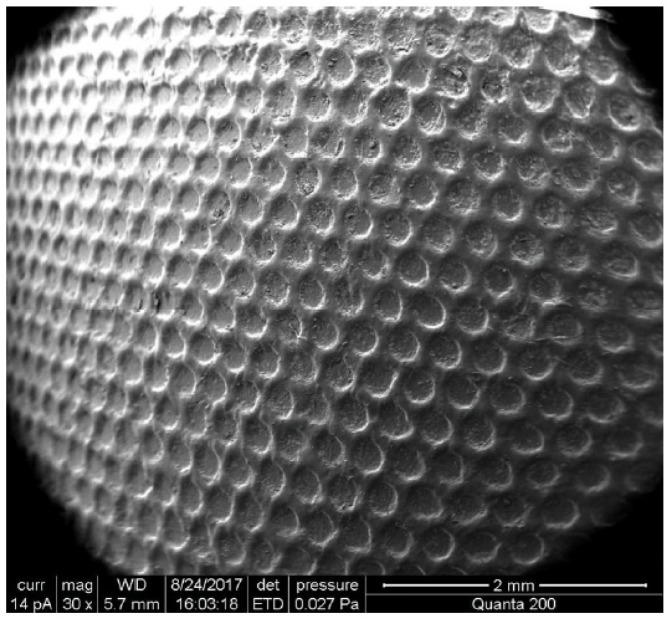
SEM image of a textured plunger surface. Each circular feature is approximately 200 μm in diameter [23].

**Figure 3 polymers-15-02858-f003:**
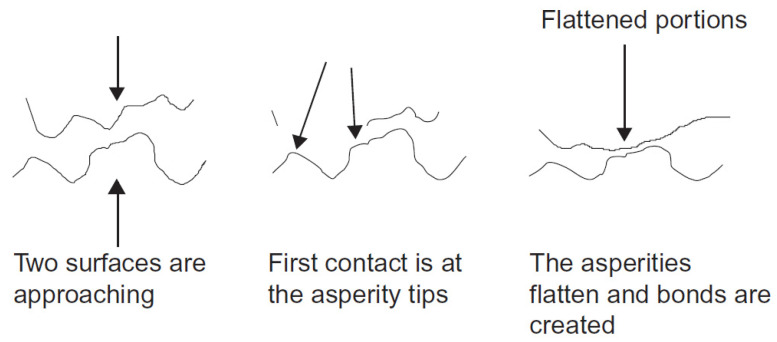
Representation of surfaces in contact [45].

**Figure 4 polymers-15-02858-f004:**
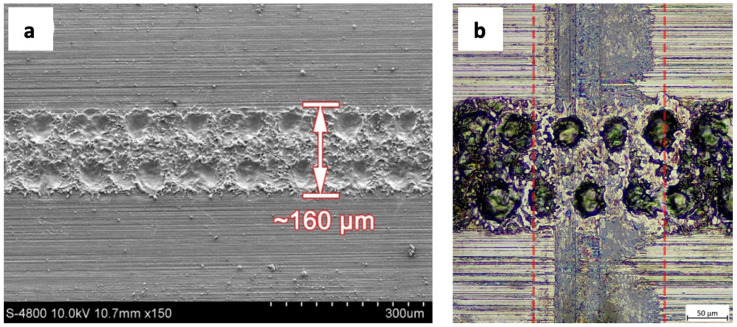
Textured surface with 63.0% dimple density, where (**a**) SEM image before sliding testing and (**b**) after sliding testing for dry conditions [86].

**Figure 5 polymers-15-02858-f005:**
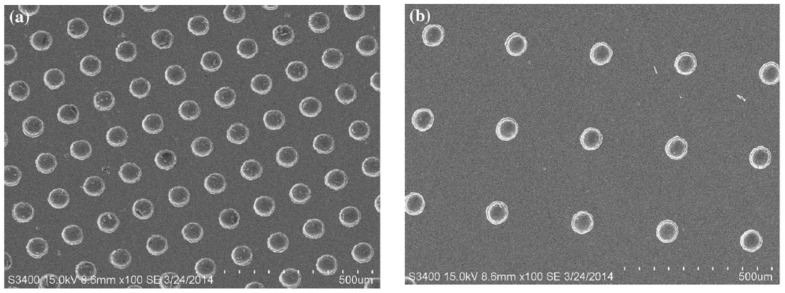
SEM images from micro dimples, where (**a**) represents 5 % density and (**b**) shows 25 % density of micro dimples [87].

**Figure 6 polymers-15-02858-f006:**
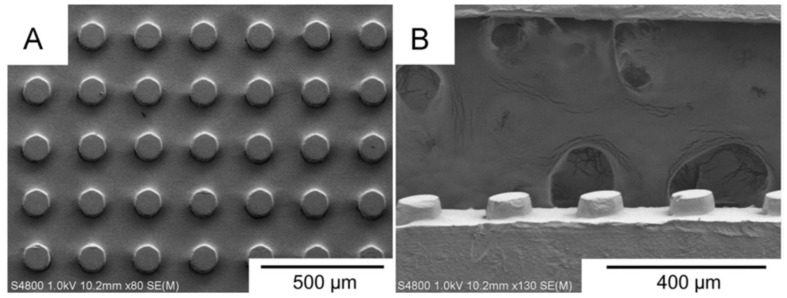
SEM images of micro injection-moulded PP samples with tapered cylindrical pillars: (**A**) plan view, and (**B**) side view [92,93].

**Figure 7 polymers-15-02858-f007:**
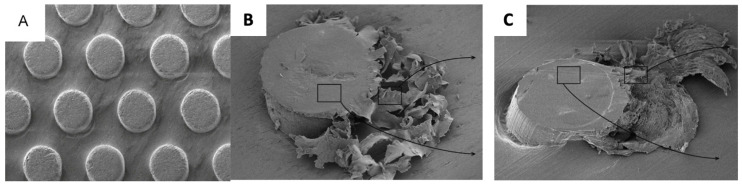
SEM images of textured POM surface showing (**A**) plan view of textured surface, and (**B**) and (**C**) worn pillars on surface [92,93].

**Figure 8 polymers-15-02858-f008:**
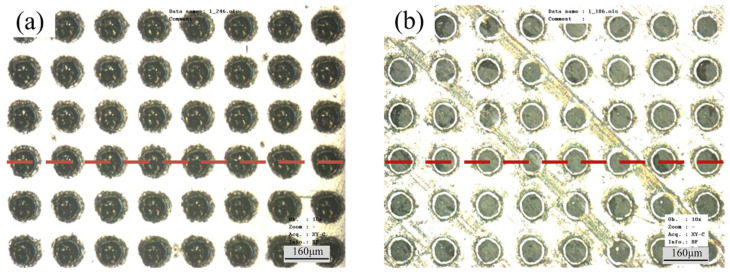
Array of circular concave features (**a**) before and (**b**) after wear testing [94].

**Figure 9 polymers-15-02858-f009:**
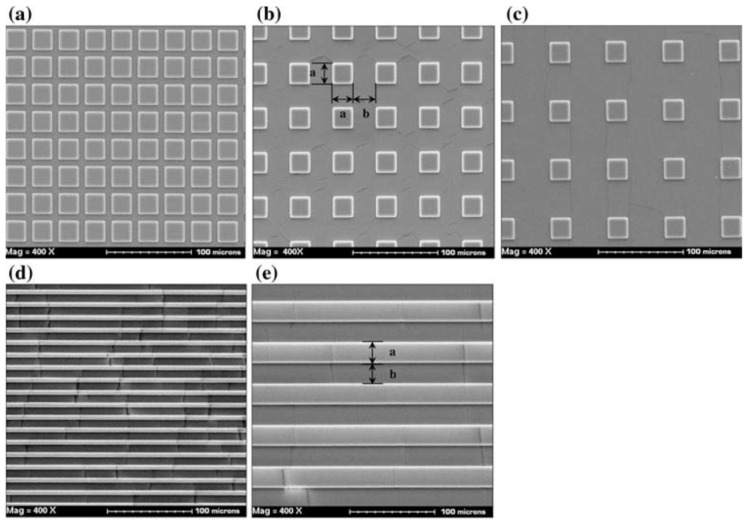
SEM images of textured PDMS produced by micro moulding where (**a**–**c**) are of square pillars with different pitch and (**d**,**e**) are of grooves with different pitch [96].

**Figure 10 polymers-15-02858-f010:**
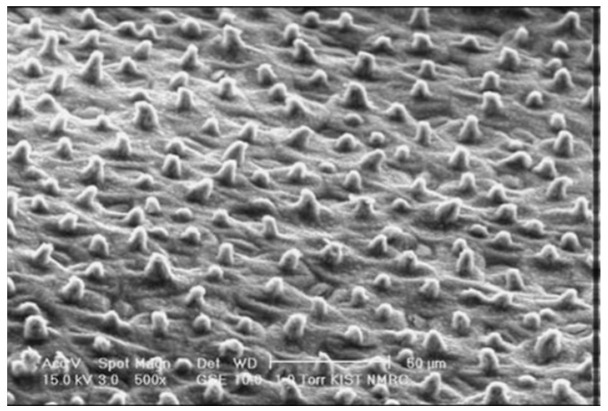
Surface of lotus flower showing super hydrophobic water-repelling protuberances [98,99].

**Figure 11 polymers-15-02858-f011:**
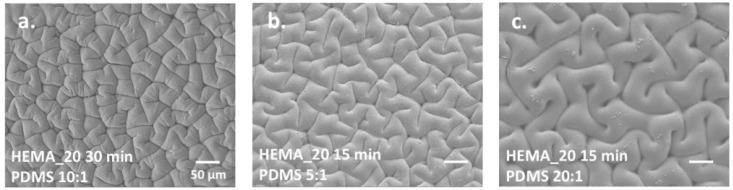
SEM images of surface morphology of PDMS modified with HEMA solutions. Images (**a**–**c**) correspond to different processing conditions [104].

**Figure 12 polymers-15-02858-f012:**
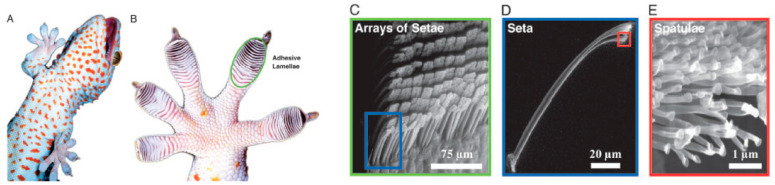
(**A**) gecko, (**B**) gecko foot (detailed), (**C**–**E**) are detailed observations of the structures present in a gecko’s foot [106].

**Figure 13 polymers-15-02858-f013:**
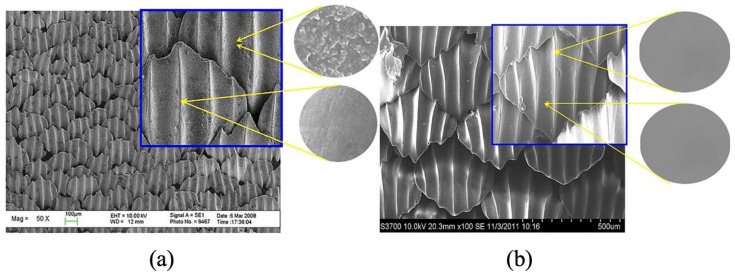
SEM images of the surface of (**a**) shark skin and (**b**) biomimetic synthetic shark skin [111].

**Table 1 polymers-15-02858-t001:** Polymer medical devices and applications.

	Polymer Types	Properties				Applications	Reference
		Biocompatibility	Mechanical Behaviour [MPa]		Sterilization Methods		
			Young’s Modulus	Tensile Strength			
General use	Polyethylene	x	(483–1750)	(2.69–200)	Cleaning Products; Autoclave; Steam	Prosthetics	[30,31]
	Polypropylene	x	(680–3600)	(16.7–45.0)		Syringes; Drug delivery systems; Packaging; Connectors; Finger joint prostheses	[30,31]
	Polystyrene	-	(2590–4710)	(34.0–55.0)	Ethylene oxide; Gamma irradiation	Light weight packaging; Absorbing impact energy	[32,33,34]
	Polyvinyl chloride	-	(3.45–73.1)	(1820–7030)	Steam; Autoclave	Disposable medical devices; Cardiac catheters; Blood bags; Haemodialysis devices	[4,35]
Engineering	Polymethyl methacrylate	-	(60.0–74.5)	(3000–3300)	E-beam; Gamma radiation; Ethylene oxide.	Endoscopic medical parts; Dental and orthopaedic surgeries	[36,37,38]
	Polycarbonate	x	(40.0–154)	(1800–6000)	Ethylene oxide; Autoclave; Steam	IV connectors used in renal dialysis; Cardiac surgery	[36]
	Polyurethane	x	(0.345–34.5)	(1.14–248)	Ethylene oxide; E-Beam; Gamma radiation	Encapsulants; Dip moulded gloves and balloons; Catheters	[39]
	Polyacetals	-	(586–11700)	(21.0–75.8)	Steam; Autoclave; Ethylene oxide	Tubs for drug delivery; Pacemakers; Prosthetics	[36,40]
	Polyesters	x	(1000–10,600)	(10.0–123)	Steam; Autoclave; E-Beam; Gamma radiation	Implants; Screws; Scaffolds; Tissue engineering; Prosthetics	[41,42]
	Polyamide	-	(3030–5520)	(62.1–122)	Autoclave; Steam	Stent delivery systems; Prescription bottles	[43]

Note: elasticity modulus and tensile strength values are representative of the range used.

**Table 2 polymers-15-02858-t002:** Reported coefficients of friction associated with various textured polymer materials.

Microstructure Geometry	Textured Material	Contacting Material	Load	Sliding Speed	Coefficient of Friction		Wear Rate		Reference
					Original	Textured	Original	Textured	
Pillars	PP	Stainless Steel	2, 5, 10 N	30 mm/s	-	0.3–0.5	$4.6–5.3\times 10^{-6}$ mm/m	$6.6–25.6\times 10^{-6}$ mm/m	[93]
	POM	Stainless Steel			-	0.1–0.2	$6.1–7.5\times 10^{-6}$ mm/m	$3.1–9.9\times 10^{-6}$ mm/m	[93]
	PDMS	304 stainless Steel	5, 10, 25 mN	1 µm/s	0.7	0.4	-		[96]
	PMMA	Borosilicate	0–80 nN	2 µm/s	0.6	0.22	-		[99]
Grooves	PVS	Glass	-	-	0.32	0.19	-		[103]
	CoCrMo	UHMWPE	5, 10, 15 N	20, 40, 80 mm/s	0.12	0.096	-		[112]
	p-doped silicon	PTFE	5, 10, 25 mN	1 µm/s	0.4	0.25	*-*		[113]
	CuSn6	PTFE	100 N	200 r/min	0.19	0.165	$8.24\times 10^{-4}\;{\;\mathrm{mm}}^{3}/\mathrm{Nm}\;$	-	[114]
	AISI 1045 carbon steel	UHMWPE	1–90 N	0.05–0.3 m/s	0.262	0.092	-		[87]
	PEEK	GCRr15	0.9, 3 N	0.157 m/s	0.37	0.35	-		[95]
	SiC	PTFE	30, 75 N	3 Hz	0.32	0.10	-		[115]
	100Cr6	PEEK	2 N	20–170 mm/s	0.253	0.155	-		[100]

## Data Availability

No new data were created or analyzed in this study. Data sharing is not applicable to this article.

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
