# Peer review of "Influence of Surface Texturing on the Dry Tribological Properties of Polymers in Medical Devices"

_polymers, 2023, doi:10.3390/polym15132858_

Round 1

Reviewer 1 Report

I reviewed the manuscript titled "Influence of surface texturing on the dry tribological properties of polymers in medical devices" authored by Evangelista et al submitted to Applications of polymers in bioengineering. I find the article to be well written where authors have reviewed the research related to reduction on friction, specifically, reducing in coefficient of friction and wear in polymer surfaces. Overall, article is well written and I agree this manuscript to be accepted. Here are my recommendations -

1. Line 196: When contact occurs between a flat surface and a rough surface, abrasion is observed via material loss from the flat surface. Two body abrasive wear is generally seen when contact occurs between soft and hard surface instead of flat and rough surface?

2. Readers would want to know at what levels of texturing do the effect of friction reduction levels out? Authors may add a paragraph after line 383 discussing this aspect of their research. 

3. Line 490 and 492 needs a reference citations.

Author Response

Response to Reviewers’ Comments

“… Line 196: When contact occurs between a flat surface and a rough surface, abrasion is observed via material loss from the flat surface.  Two body abrasive wear is generally seen when contact occurs between soft and hard surface instead of flat and rough surface.”

Response: We thank this reviewer for correcting our poorly worded sentences and we have reworded these as follows (see lines 192-195):

“Abrasive wear is differentiated as either two-body abrasive wear or three-body abrasive wear. When contact occurs between a soft surface and a hard surface, abrasion is observed in the form of material loss from the soft surface.”

“Readers would want to know at what level of texturing do the effect of friction reduction levels out? Authors may add a paragraph after line 383 discussing this aspect of their research.”

Response: The reviewer rightly asks whether we can clarify the extent to which different levels of texturing influence reductions in friction.  However, this is unknown and will be a question that our future work seeks to address.  This particular part of our paper reports on work by others (Wang) who has shown that for a nominally smooth/flat PEEK surface in contact with 304 stainless steel, the coefficient of friction was 0.37, and this reduced to 0.35 for a surface of the same PEEK material that included a textured array of 25 um dimples. Wang’s work did not proceed beyond this to answer the question that this reviewer poses.

“Line 490 and 492 needs a reference citation.”

Response: We agree and have included reference [102] to address this (see Lines 488-490). “As regards thermoplastic materials themselves, various polymers and blends of polymers are formulated to provide smaller coefficients of friction as an intrinsic material property [102]”.

Reviewer 2 Report

The article discusses in detail the possibilities of improving the mechanical properties and service life of medical devices and their parts. The authors tried to collect information about currently used materials and point out the possibilities of improving their properties, durability, and comfort for the patient.

The authors give examples of a few polymeric materials used in the healthcare industry. It would be good if the authors also mentioned the need for polymers with antibacterial treatment. A promising material could be, for example, polymers doped with silver nanoparticles. They are discussed in articles: 10.3390/polym15020379, which the authors could cite. Moreover, the addition of metal nanoparticles to the polymer matrix could certainly positively affect the tribological properties of the polymer. The authors mention antibacterial properties only marginally at the beginning of the article.

Although this article does not provide solutions in the field of developing new materials, it can still be beneficial for readers who are trying to find their way around the issue of the application of materials in the healthcare industry that is exposed to friction.

The article offers a broad overview of materials and focuses on the texturing of polymers.  I appreciate the volume of processed literature and therefore recommend this manuscript for publication.

Author Response

Response to Reviewers’ Comments

“…It would be good if the authors also mentioned the need for polymers with antibacterial treatment. A promising material could be, for example, polymers doped with silver nanoparticles. They are discussed in article: 10.3390/polym15020379, which the authors could cite. Moreover, the addition of metal nanoparticles to the polymer matrix could certainly positively affect the tribological properties of the polymer. The authors mentioned antibacterial properties only marginally at the beginning of the article.”

Response: We appreciate this reviewer’s comment and have included this additional reference.  It is now included as Ref [15].
